# Does Seasonality, Tidal Cycle, and Plumage Color Influence Feeding Behavior and Efficiency of Western Reef Heron (*Egretta gularis*)?

**DOI:** 10.3390/ani10030373

**Published:** 2020-02-26

**Authors:** Ahmed Al-Ali, Sabir Bin Muzaffar, Waleed Hamza

**Affiliations:** 1Department of Biology, United Arab Emirates University, Sharjah 40746, UAE; 2Department of Biology, United Arab Emirates University, Ai-Ain 15551, UAE; w.hamza@uaeu.ac.ae

**Keywords:** Western Reef Heron, feeding, behavior, foraging efficiency, foraging success ratio, bio-indicator, morph, mangrove habitat

## Abstract

**Simple Summary:**

Western Reef Heron (*Egretta gularis*, Ardeidae) is a widely distributed species occurring in the Arabian Gulf in which individuals are either dark or light morphed. Most (over 70%) of the population consists of dark morphed birds and it is suggested that morphological characters are influenced by feeding behavior, predator–prey relations, and the environment. We studied the feeding behavior in the two color morphs to better understand the factors that influenced feeding behavioral diversity and efficiency. We recorded 13 feeding behavior types with difference in their utilization between seasons and age groups. Stand and wait and slowly walking were the two most commonly used techniques in both morphs. Feeding behavioral diversity was higher in both morphs in summer, probably because summers are harsh and abundance of food is lower. Feeding behavioral diversity was higher in dark morphs in general and was even higher in summer at falling tides. Foraging efficiency, however, did not vary between seasons or morphs. Feeding behavioral diversity and foraging efficiency was significantly higher during lag periods of rising tides in both morphs. Thus, it appears that dark morphs could be disadvantaged in summer months and therefore be utilizing a wider variety of behaviors to acquire adequate food. This study indicates that other factors, such as predator evasion or prey avoidance, may influence feeding behavior diversity and efficiency. Further studies are needed to help explain the high abundance of dark morphs in the region.

**Abstract:**

Polymorphic traits may evolve in many species of birds, often driven by multiple environmental factors. It is hypothesized that polymorphic traits in herons could be influenced by feeding behavior. Most of the Western Reef Herons (*Egretta egularis*) (more than 70%) are of the dark morph in the United Arab Emirates (UAE). Feeding behavior and efficiency in the dimorphic Western Reef Heron was characterized in a shoreline habitat of Al-Zora Protected Area, Ajman, UAE in relation to season, tidal cycle, and color morphs. Foraging behavioral observations were made using standard focal birds during summer and winter seasons spanning entire tidal cycles. Western Reef Herons used 13 feeding behavior types with difference in their utilization between seasons and age groups. Stand and wait and slowly walking were the two most commonly used techniques in both morphs. Feeding behavioral diversity was higher in both morphs in summer, probably because summers are harsh and abundance of food is lower. Feeding behavioral diversity was higher in dark morphs in general and was even higher in summer during falling tides. Foraging efficiency, however, did not vary between seasons or morphs. Feeding behavioral diversity and foraging efficiency was significantly higher during lag periods of rising tides in both morphs. Thus, it appears that dark morphs could be disadvantaged in summer months and therefore be utilizing a wider variety of behaviors to acquire adequate food. This does not explain why there are more dark morphed birds (70%) in the population. We suggest that dark morphed birds compensate for lower feeding efficiency by increasing feeding behavioral diversity and feeding efficiency during the rising tides. Further studies are needed to evaluate the influence of prey avoidance and the choice of predators that attack herons, to better understand factors influencing the numerical dominance of dark morphs.

## 1. Introduction

The evolution of polymorphism is widespread in the animal kingdom and polymorphic traits have been shown to be of adaptive significance to many animal groups including birds [1,2]. Multiple factors could enable the evolution and maintenance of polymorphism [2]. For example, variable morphological traits could confer selective advantages in social interactions, mate selection, reproductive behavior, prey selection, predator avoidance, and crypsis [1,2]. However, experimental evidence documenting morphological traits and their adaptive advantages are limited relative to the widespread occurrence of polymorphism. It is hypothesized that discrete polymorphic traits (such as dark versus light plumage in birds as diverse as egrets, hawks (Accipitridae) and skuas (Stercorariidae)) are maintained through disruptive selection, with each morph exhibiting comparable levels of fitness, thereby allowing each morph to be sustained in populations [2,3]. 

Five species of egrets and herons (Ardeidae) have dimorphic plumage with a distinct dark and light plumage in adult birds [4,5]. Predation, prey selection, niche segregation, crypsis, and environmental conditions have all been regarded as important determinants of color polymorphism in this avian group [3,6,7,8,9]. Murton [6] hypothesized that the color of plumage was involved in determining foraging tactics and diet. Light morphs were putatively less conspicuous to fish against a bright sky while dark morphs were more visible. This could result in alteration of foraging tactics (e.g., light morphs using more passive tactics to capture fish versus active foraging tactics in dark morphs [3,6,8]). Similarly, white Pacific Reef Herons (*Egretta sacra*) used passive hunting techniques to find prey in the Cook Islands [8], whereas dark individuals hunted by actively walking or running on reefs [8]. This could also cause variations in diet due to avoidance of dark morphs by prey species. For example, mosquito fish (*Gambusia affinis*) avoid areas occupied by dark birds compared to those occupied by light birds, suggesting that light plumed birds were more cryptic [9]. Caldwell [6] provided support for an alternate scenario relating to crypsis in which white individuals of Little Blue Herons (*Egretta caerulea*) were subject to greater number of predatory attacks (from raptors) in Panama because they were more visible to the aerial predators, compared to dark individuals. Furthermore, high wind velocity, temperature, tidal phase, seasonal phase, water depth, or the presence of shoreline vegetation (such as mangroves) could all result in altered patterns of foraging that differentially benefits one morph over the other [3,8,9,10,11,12]. Interestingly, it is sometimes evident that none of the factors adequately explain the relative distribution of alternate morphs of a species across its geographic range. For example, it is not clear why the Eastern Reef Heron (*Egretta sacra*), distributed in eastern Asia, has more white morphs moving away from Japan towards Australasia [11]. All of these factors seem to exert selective pressures over these morphs and their relative importance in determining the frequency of each morph in a given population needs to be better studied and elucidated [2].

The southern shoreline of the Arabian Gulf serves as an important stopover and breeding area for large populations of waterbirds [13]. A substantial portion of these areas fall within the territory of the United Arab Emirates (UAE) that consists of a combination of mudflats, seagrass beds (*Halodule universis*, *Halophila ovalis*, and *Halophila stripulacea*) and mangrove patches (*Avicennia marina*) separated by sandy beaches. One of the most abundant waterbird species in the UAE is the Western Reef Heron (*Egretta gularis*) which is regarded as a coastal resident and inland visitor [13,14,15] with an estimated 500–1000 pairs breeding in UAE and an estimated 3000 pairs breeding throughout Arabia [13,15]. It is a dimorphic, medium-sized heron occurring usually in salt and brackish waters throughout the coastal zone [13,15]. There are two distinct morphs, one being a slaty gray with a small white gular patch and the second being white [4]. Both morphs have been recorded in coastal areas, mudflats and mangroves and rarely in inland locations [13,15]. The abundance of individuals exhibiting the white morph ranges from 10% to 50% of the total population [4,13,15] and this varies by region. For example, in Bahrain and Pakistan only 19–25% of the population consists of white morphs [4,13,15,16]. Three subspecies are recognized for Western Reef Heron with *Egretta gularis schistacea* occurring in UAE and spanning in distribution from the East African coastline, Red Sea, Arabian Gulf, and West, South, and South East India, and Sri Lanka [4,13].

The UAE is a country with a hyper-arid environment, where the marine and shoreline environments are valuable due to the ecosystem services they provide and the marine invertebrates, fishes, and avifauna that they support [17]. Wetlands can be easily impacted by anthropogenic stressors as they attract people for fishing or leisure activities or developers converting wetland areas to lodges or hotels [18,19]. Better understanding of the factors influencing ecology, feeding preferences, and foraging sites of waterbirds could direct conservation efforts of wetlands.

The objectives of this study were (1) to describe feeding behavioral diversity of the Western Reef Heron; (2) to determine the influence of tidal cycle and seasonality on feeding behavior and efficiency of the two morphs; and (3) to determine the influence of color morphs of the Western Reef Heron on the diversity of feeding behavior types and the foraging efficiency.

## 2. Materials and Methods

The study was conducted with permission from the site authority in Al-Zora protected area, Ajman Emirate, UAE (Figure 1) that is managed by the Ajman Municipality. The protected area is listed as a Ramsar site [20] and consists of mangroves and salt marshes with a total area of 1.95 km^2^. We chose this site due to its limited human disturbance and the high abundance of the Western Reef Heron reported in Ebird.org. The study site was within the eastern side of protected area boundaries which contained shallow permanent marshes and mudflats influenced by the tide (Figure 2). We used a total of 70,909 m^2^ split evenly into eight quadrates that were 8863 square m^2^ each.

We assessed foraging activity to cover two seasons (summer and winter) to evaluate the influence of high verses low temperatures on feeding [21,22]. We used the temperature data from Ajman between 2007 and 2016 from the National Center of Meteorology in the United Arab Emirates [23]. The highest temperatures were recorded in August while lowest temperatures were recorded in December and January [23]. Thus, the months of August 2017 and December 2017–January 2018 were chosen to represent summer and winter seasons, respectively. 

Throughout the month in each season, observations were recorded every other day to account for changes in the tidal phases within one lunar cycle. Different behavioral types were categorized and an ethogram was created based on [24,25] including the following behaviors: Stand and Wait, Bill Vibration, Standing Flycatching, Walking Slowly, Walking Quickly, Running, Hopping, Leapfrog Feeding, Open-Wing Feeding, Canopy Feeding, Foot Raking, Dipping, Feet First Diving, with subdivisions in each behavior type (described in Table 1). We selected an actively feeding bird (classified as focal bird, 25) for observations. The feeding behavior types of the focal bird was recorded systematically during a 1-min observation period at 4-min intervals [24,25]. Each observation was recorded as one or more behavior types (either ‘Stand and Wait’, ‘Bill Vibration’, etc.) and each recorded behavior constituted a statistically independent data point over the course of a 1-min period [24,25]. The morph of the bird was also recorded. Observations started at 07:30 h and ended at 12:30 h (that is, a total duration of 5 h). All the data were recorded into a voice recorder at the time of observation. The date, time, and observation number were clearly stated into the voice recorded prior to the 1-min observation period. Once the observation period started, feeding behavior types (either ‘Stand and Wait’, ‘Bill Vibration’, etc.) were clearly stated into the recorder as they were observed. Once the period ended (that is, during next the 4-min interval), the recorder was stopped and notes were written on the field log sheet while waiting for the next observation period. The tidal cycle was recorded as ‘rising’ tide (defined as increasing water before reaching high tide) or ‘falling’ tide (defined as receding water levels after the high tide), based on the known timing of the tidal cycle on a given day.

A total of 15 days of observations were recorded in winter and summer, with 61 independent observations each day and 915 records each season, totaling to 1830 records for the whole study period. Observations were made from a fixed vantage point in the eastern portion of the study area (to cover the whole area while maintaining a good view of the birds) using an optical telescope (Kowa TS-611 Spotting Scope 60mm with 27X Eyepiece, Kowa Optics, Duesseldorf, Germany) and binocular (Nikon 12 × 42 Binocular, Tokyo, Japan). Whenever activity was present, defined by walking around searching for prey [21] or actively feeding from Stand and Wait positioning (Table 1), the observation was recorded vocally using a sound recorder. Basic information (such as weather patterns including windy, sunny, foggy, etc.) and some relevant field notes were recorded in the field log sheet and the voice notes were transcribed later. A DSLR camera with a Telephoto lens (Canon 7d Mark ii with 600mm Lens, Tokyo, Japan) was used whenever needed.

The Shannon’s diversity index *H* for behavior (number of different behavior types and their evenness) was calculated using software PAST (Paleontological Statistical Software, University of Oslo, Oslo, Norway) using the following formula:(1)$$H\;=\;-\;\sum_{i}^{S}pi_{\;}\log\;p_{i}$$
where:
*S* = the total number of behavior types;*i* = the number of observations of each behavior type; and *p_i_ =* relative proportion of each behavior type. 

Statistical comparisons of Shannon’s *H* were done using the Diversity t test that utilizes Monte Carlo randomization methods. 

Number of strikes (attempts to capture prey) and number of successful strikes (successful feeding attempts) were recorded for selected birds. We then calculated the foraging efficiency (FE) by dividing successful strikes by the total number of strikes using the following formula following Kuranchie [22]
FE = (Successful Strikes)/(Total Strikes) × 100(2)

Foraging efficiency of each age group (juvenile, adult, breeding adult) was calculated using the following formula:FE Age Groups = (FE age group)/n(3)

We used the Mann–Whitney U test to examine FE differences between seasons, tidal phases and morphs, as the data did not fit assumptions of normality. Statistical analyses of foraging efficiency were done using IBM SPSS Statistics (Version 25) statistical software (1989–2017, New York, NY, USA). The significance value was set at 0.05 for all statistical tests. 

## 3. Results

A total of 338 out of 1830 observations had Western Reef Heron present in the study area (18.5%). Out of these 338 observations, 114 observations were of feeding birds (33.7%). Approximately 79% of the Western Reef Herons in our study were dark morphed while the remaining 21% were white and this pattern was consistent between seasons. Western Reef Heron individual records in summer season summed up to 148 with highest single day count of 29 individuals. In contrast, winter season records had a total count of 479 with highest single day count of 57 individuals. Western Reef Herons were occasionally seen feeding in a group. Seventeen species were observed to feed close to feeding Western Reef Heron. The majority of Western Reef Heron (>80%) were found to be feeding while they were wading in the water and very few individuals were seen feeding while walking on salt marsh habitats. One heron on a high tide was observed snatching insects from mangroves. In summer, the earliest feeding was at 07:30 h and the feeding rates increased by 07:40 h while in winter the earliest feeding activity started at 07:40 h and increased slowly by around 08:10 h. Once feeding was increased, clear temporal patterns of high or low feeding activity were not evident.

### 3.1. Feeding Behavior Types

Thirteen feeding behaviors were used by the Western Reef Heron out of the 28 previously recorded feeding behaviors of egrets and herons in general. Adult birds used 11 feeding behaviors which were stand and wait, walking slowly, foot raking, walking quickly, open-wing feeding, bill vibration, hopping, leapfrog feeding, canopy feeding, dipping, feet first diving. Sub-adult birds used four feeding behaviors: stand and wait, walking slowly, foot raking, walking quickly in descending order. Both sub-adult and adult birds used the same descending order of behavior usage which were stand and wait, walking slowly, foot raking, walking quickly. 

### 3.2. Seasonal Effects, Tidal Influence, and Inter-Morph Differences

Diversity of behaviors varied significantly between seasons, with a significantly higher diversity noted in summer (Shannon’s *H* = 1.78) compared to winter (Shannon’s *H* = 1.41, *t* = 5.72, *p* < 0.00001, Figure 3, Table 2) for all birds. There was no difference between the diversity of behaviors observed between rising and falling tide periods across seasons (Shannon’s *H* = 2.22 at rising tide and *H* = 2.08 at falling tide, *t* = -1.66, *p* > 0.05). In general, dark morphed herons had significantly higher diversity of behaviors (Shannon’s *H* = 1.68) compared to lighter herons (Shannon’s *H* = 1.54, Diversity *t*-test, *t* = 2.07, *p* = 0.039), irrespective of season. When tides were separated, dark morphed birds had significantly higher diversity of behaviors (*H* = 1.78) compared to light morphed birds (Shannon’s *H* = 1.56, Diversity *t*-test, *t* = 2.03, *p* = 0.04, Table 2) during falling tides in the summer only (Table 2).

### 3.3. Foraging Efficiency

Foraging efficiency appeared to be 4% higher in the winter season although this was not significantly different (*p* > 0.05, Table 3). When data from both seasons were pooled, breeding adults, non-breeding adults and sub-adults had similar FE (16.67–18.18%), while juveniles had the lowest FE (6.25%), although this was not tested statistically. Foraging efficiency was significantly influenced by the tidal cycle, with birds having greater FE when the tide was rising (*p* = 0.025, Table 3). Foraging efficiency was not significantly different after controlling for season and tidal effect between the two morphs (*p* > 0.05, Table 3), although the mean value of FE was higher in light morphed birds.

## 4. Discussion

The summers in UAE are extreme, with temperatures exceeding 50 °C, and regular dust storms and almost no precipitation [13]. As a result, a large number of coastal waterbirds migrate to areas with cooler temperatures and better food resources [13]. The generally improved conditions in winter, when temperatures become mild, attracts migratory as well as resident birds to coastal areas including mangroves [13]. Western Reef Herons disperse widely at times of low food abundance or harsh weather and are known to return to areas with high food abundance when weather patterns improve [15]. 

### 4.1. Behavioral Diversity

The most frequently recorded behaviors were stand and wait and walking slowly although many other behaviors were observed. This is consistent with the known visual acuity and efficiency of striking of Western Reef Herons [26]. Western Reef Herons typically have high levels of visual acuity that corrects for refraction of light when attacking submerged prey. The hunting posture is split into a ‘pre-strike’ position where the head is arched at an angle of 60°. This is then followed by the actual ‘strike’ at high velocity (up to 270 cm/s) which results in capture of prey [26]. Furthermore, many species of egrets and herons exhibit stand and wait or walking slowly as foraging tactics. Tojo [27] observed both behaviors (stand and wait and walking slowly) in Grey Herons (*Ardea cinerea*), Intermediate Egrets (*Egretta intermedia*), while mostly walking slowly in Great White Egrets (*Egretta alba*) and Little Egrets (*Egretta garzetta*). Similarly, Great Blue Herons (*Ardea herodias*) also showed both behaviors in adults and juveniles [28]. In comparison, Cattle Egrets (*Bubulcus ibis*) used stand and wait behavior in farmlands [27] while Little Blue Herons (*Egretta caerulea*) used primarily walking slowly as a foraging tactic [29]. Thus, Western Reef Herons seem to conform with passive hunting techniques used in herons and egrets [24].

### 4.2. Behavioral Diversity and Foraging Efficiency

The extreme weather of summers in the Arabian Peninsula makes food acquisition difficult [13]. Thus, we expected that overall feeding behavioral diversity would be higher in summer, when individual birds would have to employ a wide range of feeding tactics to find food. Conversely, lower diversity of behaviors in winter could be linked to abundant food and better temperatures allowing large aggregations to feed with relatively less effort, employing fewer foraging tactics. 

More Western Reef Herons were present (feeding or non-feeding) in larger numbers during the rising tide phase than during falling tide. Although the diversity of feeding behaviors did not vary with the tidal cycle, FE was higher during the rising tide. This is consistent with other studies relating to tidal effects on FE in herons and bitterns. For example, Grey Herons increase in numbers at low tide (rising tides) and decrease in numbers at high tide, presumably taking advantage of inter-tide lag periods [11]. This pattern is observed in other herons and in other shorebirds, such as Sanderlings (*Calidris alba*) [30,31]. It is suggested that tidal lag periods aid in exposing the maximum amounts of small prey from the benthic communities as well as force small fishes into shallower areas [11,24,30,31], allowing shallow water coastal foragers to take advantage of this increase in food, as observed in Western Reef Herons. The finding that dark morphed birds used a higher diversity of behaviors to find food in summer during falling tide is interesting. We suggest that dark morphed birds could be disadvantaged in relation to crypsis [3,6,9] and therefore had to employ a greater diversity of behaviors to successfully capture food in the summer. 

Foraging behavior utilization between different age-groups showed increasing diversity of behaviors as bird age increased. Foraging behavior may be transmitted over generations as young birds learn from their parents or from social settings and subsequently transmit what they learned to their offspring later in life [32]. Great Egrets in solitary groups performed poorly compared to those feeding in large social groups [33]. The fact that juvenile birds exhibited a lower number of behavior types compared to sub-adult birds and adults might suggest that social learning during foraging may occur in Western Reef Heron [32,33].

### 4.3. Foraging Efficiency

Juveniles had lower FE compared to sub-adults and adults possibly because age is often related to experience and younger or non-breeding birds often have less experience in hunting [24]. In contrast, the breeding adults are considered to be more experienced and therefore better at catching prey [24]. This might be because of the nutritional needs for mating or parental care supply as breeding occurs in summer in Western Reef Herons in the Middle Eastern region [13] and this is associated with more energy invested in preparation for breeding [24].

Foraging efficiency in the Western Reef Heron was generally lower than other species. For example, Kent [28] showed that Little Blue Heron, Snowy Egret (*Egretta thula*), Tricolored Heron, (*Egretta tricolor*) had 59.0%, 42.8%, and 33.0% foraging efficiencies, respectively, compared to the 17.30% in summer and 13.82% in winter for Western Reef Herons in this study. Some of these differences could be attributed to inter-species differences [24,29]. Each of the species above have slightly different choices of food and they effectively partition their habitat to avoid resource competition [29]. In addition, FE of Great Blue Herons changes with group size, with birds feeding in larger flocks having higher efficiency [28]. We did not measure FE in relation to group size in the Western Reef Herons in the current study and this needs to be investigated in the future. We suggest that the generally low visibility due to the high sediment content of the water could lead to low FE in Western Reef Herons.

White morphs of Western Reef Herons in our study did not differ significantly in FE from dark morphs. In the Cook Islands, white morphed Pacific Reef Herons used aerial hunting techniques to find prey [8]. In comparison, dark morphs of Pacific Reef Herons hunted using walking or running on reefs [8], comparable to the most commonly used feeding strategy observed in our study. We did not measure other factors, such as predator preference and avoidance of dark morphs by the prey which could differentially influence crypsis in dark and light morphed birds [9]. We suggest that dark morphs are likely to blend more with the dark background of mangrove areas (due to vegetational complexity) or against foggy or cloudy skies, consistent with other studies [9]. Our study is unable to explain why dark morphed birds are more abundant, and studies examining crypsis involving predator avoidance or prey avoidance of different morphs may shed more light on this matter [9]. 

The observation of a larger proportion of dark morphed birds remains enigmatic and is contrary to the assumption that the white morph probably dominates in other Arabian populations [13]. It appears that the white morph individuals are typically less abundant in Bahrain, North Egypt, and Pakistan (19–25%) [15] and less than 1% in Senegal, Gambia, and Mauritania populations [4]. We suggest that white morphed individuals are dominant in *E. g. gularis* while dark morphed birds are the dominant morph in *E. g. schistacea*. Further studies are needed to better understand the relative abundance of each morph especially for the eastern populations. Further studies examining evasion of herons by prey species (such as fish) and targeting of herons by predators (such as hawks) could help better explain the relative proportions of each morph.

## 5. Conclusions

We conclude that diversity of feeding behaviors in Western Reef Herons was comparable to other species of herons and egrets. Thirteen different behavior types were used and stand and wait and slowly walking were commonly used behavior types. Feeding behavioral diversity was higher is summer, suggesting that harsh conditions required a variety of feeding behavior types to result in success. Feeding behavioral diversity was higher in dark morphs in general and was even higher in summer during falling tides. We suggest that dark morphs could be disadvantaged during the summer. Feeding efficiency did not differ between seasons or morphs. Feeding behavioral diversity and foraging efficiency was significantly higher during lag periods of rising tides in both morphs. Multiple factors could be influencing feeding behavior and efficiency in Western Reef Herons and further studies are needed to tease out the role of predator avoidance or evasion by prey in determining feeding strategies.

## Figures and Tables

**Figure 1 animals-10-00373-f001:**
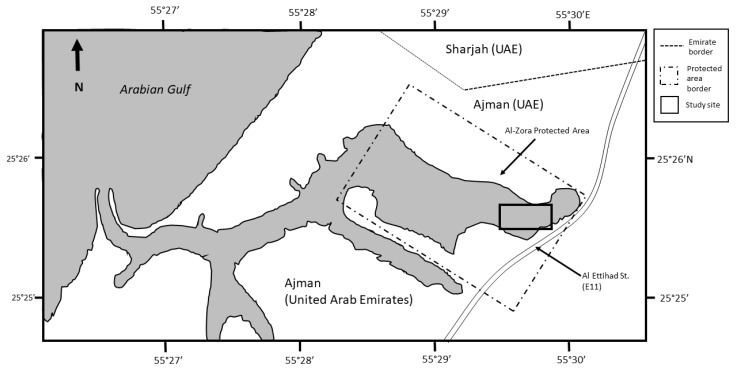
Map of Al-Zora Protected Areas, Ajman, UAE.

**Figure 2 animals-10-00373-f002:**
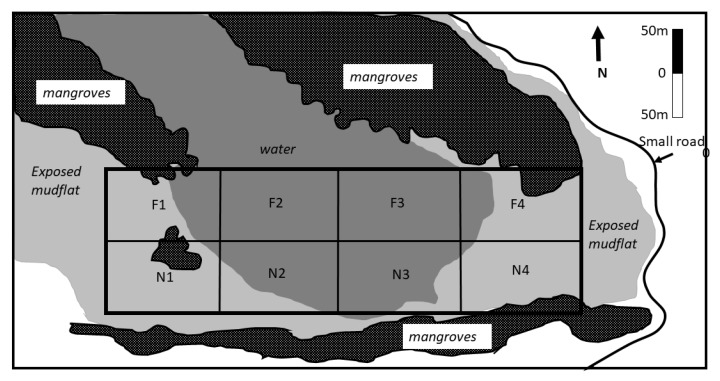
Spatial distribution of habitat types and allocation of grids used to record observations of feeding birds.

**Figure 3 animals-10-00373-f003:**
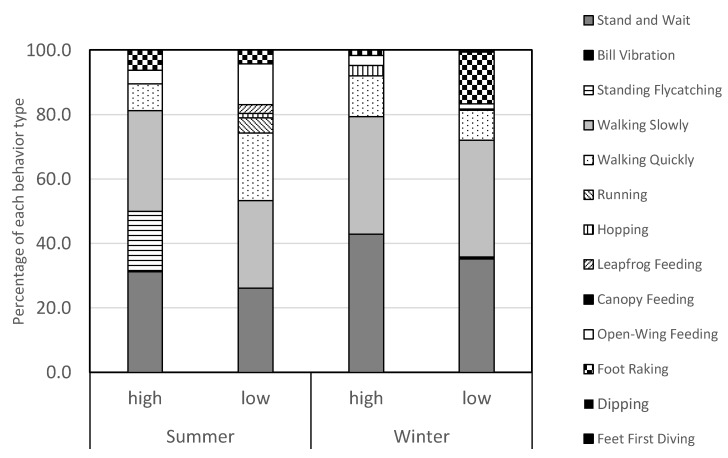
Variation in the proportion of each behavior in relation to seasons and tidal cycle. All behavior types observed in extremely low frequencies (<0.70%) are pooled together and designated ‘black’ colors.

**Table 1 animals-10-00373-t001:** Ethogram of behavior types in Western Grey Herons observed in this study (from Kushlan 1976 [24]).

Category and Number	Behavior Type
**A**	**Stand (Stalk)**
1	Stand and Wait: the bird stands still and waits for prey to approach
2	Bill Vibration: the bird places its bill in the water and rapidly opens and closes it to attract prey
3	Standing and Flycatching: bird remains standing and catches flying insects
4	Walking slowly: the bird walks slowly in shallow water, searching for prey
**B**	**Disturb and Chase**
5	Walking quickly: the bird walks quickly, causing disturbance which forces prey to emerge
6	Running: the bird runs to chase specific prey, often using the wings for balance and uplift
7	Hopping: the bird flies to a feeding site a short distance away
8	Leapfrog feeding: the bird flies from the rear to the front of a forward moving feeding flock
9	Open wing feeding: the bird extends one wing and keeps it extended for a few seconds before retracting it
**C**	**Aerial and Deepwater Feeding**
10	Canopy Feeding: the bird extends both wings like an umbrella with wing-tips touching the water and keeps its head underneath the wings to strike prey
11	Foot Raking: the bird drags its toes across the substrate to disturb prey
12	Dipping: the bird flies along and periodically dips into the water to pick up prey
13	Feet First Diving: the bird plunges from the air into the water feet first to strike prey

**Table 2 animals-10-00373-t002:** Comparison of diversity of feeding behaviors (Shannon’s *H* ± variance) in Western Reef Herons in relation to season, tidal cycle, and morph type.

Statistical Test	Summer	Winter	*t*	Df	*p*
Shannon’s *H* (all birds)	1.79 ± 0.002	1.42 ± 0.002	5.72	591	<0.00001
	Rising tide	Falling tide			
Shannon’s *H* (all birds)	2.22 ± 0.001	2.08 ± 0.006	−1.66	165	0.090
	Dark morph	Light morph			
Shannon’s *H* (morphs	1.68 ± 0.002	1.53 ± 0.003	2.07	467	0.039
Shannon’s *H* (summer, falling tide)	1.78 ± 0.003	1.56 ± 0.008	2.03	101	0.044

**Table 3 animals-10-00373-t003:** Comparison of the foraging efficiencies (FE) in percentage (± standard error estimated from randomizations) of Western Reef Herons in relation to season, tidal cycle, and morph.

Feeding Efficiency in Relation to Different Variables	Summer	Winter	Mann-Whitney U	z-Score	*p*
FE_season_	17.30 ± 4.97	13.82 ± 3.48	53.0	−0.0356	0.9716
	Rising	Falling			
FE_tides_	20.08 ± 4.99	7.58 ± 2.19	33.0	−2.2409	0.0250
	Dark	Light			
FE_morph_	12.02 ± 2.28	20.85 ± 6.57	41.5	−0.85447	0.3929

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
