# Peer review of "Does Seasonality, Tidal Cycle, and Plumage Color Influence Feeding Behavior and Efficiency of Western Reef Heron (Egretta gularis)?"

_animals, 2020, doi:10.3390/ani10030373_

Round 1

Reviewer 1 Report

Revision is much improved. Only minor edits. 

Line 15: change to "understand the factors that influenced..."

Line 23: recommend changing to "dark morphs could be disadvantaged..." Same with Line 41 (insert "be")

Line 46: Recommend change to "influence of prey avoidance and the choice..."

Line 47: I am not sure "dominance" is best term here, the morph is dominant in numbers (ratio) but this term could be misconstrued to mean one morph is dominant (behaviorally) over the other

Fig 1 and Fig 2 require a "north arrow" and scale bars. 

Line 178: Recommend change to "examine FE differences between seasons..."

Line 180: Remove one of the "the" in last sentence.

Line 193 and 196: recommend using "temporal" in front of pattern to clearly define what the authors mean by feeding activity

Line 267-270: Unclear what the authors are trying to state here. Cattle Egrets and Little Blue Heron use a variety of feeding behaviors and the way this is currently it written, it implies that Little Blue Heron use only "walk slowly" as a foraging tactic which is not correct. For other herons, I think they have generalized too much as well in these sentences. Kushlan (24) and Kushlan and Hancock (26) both describe the wide variety of feeding behaviors for many species. 

Author Response

Revision is much improved. Only minor edits. 

Line 15: change to "understand the factors that influenced..."

Response: Done

Line 23: recommend changing to "dark morphs could be disadvantaged..." Same with Line 41 (insert "be")

Response: Done

Line 46: Recommend change to "influence of prey avoidance and the choice..."

Response: Done

Line 47: I am not sure "dominance" is best term here, the morph is dominant in numbers (ratio) but this term could be misconstrued to mean one morph is dominant (behaviorally) over the other

 Response: Done

Fig 1 and Fig 2 require a "north arrow" and scale bars. 

Response: These were done in response to comments of the 1st to reviewers.

Line 178: Recommend change to "examine FE differences between seasons..."

Response: Done

Line 180: Remove one of the "the" in last sentence.

 Response: Done

Line 193 and 196: recommend using "temporal" in front of pattern to clearly define what the authors mean by feeding activity

 Response: Done

Line 267-270: Unclear what the authors are trying to state here. Cattle Egrets and Little Blue Heron use a variety of feeding behaviors and the way this is currently it written, it implies that Little Blue Heron use only "walk slowly" as a foraging tactic which is not correct. For other herons, I think they have generalized too much as well in these sentences. Kushlan (24) and Kushlan and Hancock (26) both describe the wide variety of feeding behaviors for many species

Response: We agree. We wanted to state that the most common behaviors were ‘walk slowly’. We have added the word “primarily walking slowly..” to clarify this point.

We thank the last reviewer for suggesting these corrections on these minor details.

Ahmed Al-Ali

Sabir Bin Muzaffar

Waleed Hamza.

Reviewer 2 Report

An improved paper and one that provides some novel findings on the behaviour of this species of heron. A few areas for suggested improvement.

I still feel the methods are hard to follow and you need to work on how you present the activity budgets of the birds you observed to be clearer in what proportion of time was spent on different feeding activities. 

Please ensure that you thoroughly spell and grammar check this paper to remove errors and ensure quality readability.  Simple summary Line 11: put in the animal's scientific name. "The western reef heron..." Abstract Line 30: "Most (over 70%) of..." Introduction Line 60: provide the taxonomic group for skuas and hawks  Line 103-110: I like the conservation application here but it feels forced. Can you simplify to say that more knowledge on heron feeding ecology and preferences for feeding sites will allow for better direction of conservation efforts in wetlands of importance for migration waterbirds. Methods Permission / copyright for Figure 1? Same for Figure 2?

This is confusing

"Observations on focal birds were recorded in a window of 5 hours as 1-minute observations in every 5-minute interval [24, 25]. During this 1-minute observation period, one or more actively feeding birds were selected and the feeding behavior type each individual were engaged in recorded as an independent data point, along with its morph."

Do you mean:

"Focal birds were followed for behavioural recording. For a five-hour observation session, any heron actively engaged in feeding behaviour was recorded for a five minute period, with behaviour noted each minute. At the end of this five minutes, a new bird was selected for recording."

I still find the actual explanation of behavioural recording hard to follow. 

Line 164: please state the Shannon index formula and explain what each part means. Results  Line 180: the the - check spelling.  Line 183: "A total of 1830 observations were attempted, with 338 observations providing behavioural data on heron foraging activity." Table 1 is an ethogram and needs to be in the methods and not in the results. In the ethogram, please only define and describe behaviours that you actually saw and have data on. 209-end of paragraph: Seems to repeat the ethogram. Also is not results, but methods. You don't need this if you only present behaviours seen in your ethogram. Present demographic differences in behaviour in your discussion or provide data to show why differences may occur in your results.  Figure 3: Is this a graph of averaged data? If so you cannot use a stacked bar chart. You must use a column chart that allows you to include the mean or median for each behaviour. It is very hard to deduce what each behaviour is on the bars. Please re-draw to add clarity.  Discussion  Line 250: How so extreme? Line 294: You can't use less behaviour. You mean that juvenile birds have a more limited repertoire of foraging behaviours to choose from.  Line 333: Not sure that hawks are the best example here? Perhaps consider the feeding strategies of other herons or storks (maybe) as a comparison? 

Author Response

Comments and Suggestions for Authors

An improved paper and one that provides some novel findings on the behaviour of this species of heron. A few areas for suggested improvement.

I still feel the methods are hard to follow and you need to work on how you present the activity budgets of the birds you observed to be clearer in what proportion of time was spent on different feeding activities.

Response: We have elaborated on the methods.

Please ensure that you thoroughly spell and grammar check this paper to remove errors and ensure quality readability. 

Response: We have done a thorough spell and grammar check.

Simple summary Line 11: put in the animal's scientific name. "The western reef heron..."

Response: Done.

Abstract Line 30: "Most (over 70%) of..."

Response: Done.

 Introduction Line 60: provide the taxonomic group for skuas and hawks 

Response: Done.

Line 103-110: I like the conservation application here but it feels forced. Can you simplify to say that more knowledge on heron feeding ecology and preferences for feeding sites will allow for better direction of conservation efforts in wetlands of importance for migration waterbirds.

Response: Yes, we agree. We have simplified the text as suggested.

Methods Permission / copyright for Figure 1? Same for Figure 2?

Response: We have replaced them with black and white figures with the necessary details. These are our own, so the copyright is ours.

This is confusing

"Observations on focal birds were recorded in a window of 5 hours as 1-minute observations in every 5-minute interval [24, 25]. During this 1-minute observation period, one or more actively feeding birds were selected and the feeding behavior type each individual were engaged in recorded as an independent data point, along with its morph."

Do you mean:

"Focal birds were followed for behavioural recording. For a five-hour observation session, any heron actively engaged in feeding behaviour was recorded for a five minute period, with behaviour noted each minute. At the end of this five minutes, a new bird was selected for recording."

Response: NO. This is not what we mean.

We mean, a bird was chosen at the beginning of the observation period. It’s behavior was recorded every 5 minutes (NOT for 5 minutes). This occurred during a 1-mintue interval. So the data sheet would look like this 730- stand and wait, 735-stand and wait, 740-fly catching, 745-walking quickly…etc. Each datum is an independent record of a described behavior observed at the end of a 5-minute period. We have reworded the text to clarify this as follows

“We selected an actively feeding bird (classified as focal bird, 34) for observations. The feeding behavior types of the focal bird was recorded systematically during a 1-minute observation period at four-minute intervals [23, 34]. Each observation was recorded as one or more behavior types (either ‘Stand and Wait’, ‘Bill Vibration’, etc.) and each recorded behavior constituted a statistically independent data point over the course of a 1-minute period [23, 34]. The morph of the bird was also recorded. Observations started at 0730 h and ended at 1230 h (that is, a total duration of 5 hours). All the data were recorded into a voice recorder at the time of observation. The date, time and observation number was clearly stated into the voice recorded prior to the one-minute observation period. Once the observation period started, feeding behavior types (either ‘Stand and Wait’, ‘Bill Vibration’, etc.) were clearly stated into the recorder as they were observed. Once the period ended (that is, during next the four-minute interval), the recorder was stopped and notes were written on the field log sheet while waiting for the next observation period.”

I still find the actual explanation of behavioural recording hard to follow.

Response: This is standard practice for behavioral studies involving activity budgets. We refer to Altmann 1974, who first described quantification of behavior of animals that then become the standard norm for empirical analyses of behavior. Further details are provided in this reference. We also referred to Kushlan, who modified minor details for the specific case of herons and egrets. We hope the current form is clearer. (see above)

Line 164: please state the Shannon index formula and explain what each part means.

Response: We have stated the index formula. We also corrected an error where we referred to Shannon’s D as Shannon’s H. The diversity index has a symbol of D.

 Results  Line 180: the the - check spelling.  Line 183: "A total of 1830 observations were attempted, with 338 observations providing behavioural data on heron foraging activity."

Response: We would like to retain the original wording as that is the standard practice. The total number of observations could contain other behaviors, hence, depending on the study, observations of interested are listed as a subset of the total. So, 338  feeding behavioral observations were recorded out of 1830 observations in total.

Table 1 is an ethogram and needs to be in the methods and not in the results. In the ethogram, please only define and describe behaviours that you actually saw and have data on.

Response: Done

209-end of paragraph: Seems to repeat the ethogram. Also is not results, but methods. You don't need this if you only present behaviours seen in your ethogram. Present demographic differences in behaviour in your discussion or provide data to show why differences may occur in your results.

Response: Done

 Figure 3: Is this a graph of averaged data? If so you cannot use a stacked bar chart. You must use a column chart that allows you to include the mean or median for each behaviour. It is very hard to deduce what each behaviour is on the bars. Please re-draw to add clarity.

Response: This is the most common form of presenting the diversity of behavior types, expressed as proportions of each behavior type. These are not mean values. Statistical comparisons are done separately, and those are referred to in the text and in tables. The purpose of this chart is to show which behaviors were more common and which were less, in relation to tides and seasons.

 Discussion  Line 250: How so extreme?

Response; We have added the following: “..with temperatures exceeding 50°C, and regular dust storms and almost no precipitation [12]”.

 Line 294: You can't use less behaviour. You mean that juvenile birds have a more limited repertoire of foraging behaviours to choose from.

Response: Yes. WE have edited the text here.

 Line 333: Not sure that hawks are the best example here? Perhaps consider the feeding strategies of other herons or storks (maybe) as a comparison?

Response: We meant targeting of herons BY the hawks. We have edited the text to reflect this as follows: “Further studies are needed to better understand the relative abundance of each morph especially for the eastern populations. Further studies examining evasion of herons by prey species (such as fish) and targeting of herons by predators (such as hawks) could help better explain the relative proportions of each morph”

Both reviewers suggested that we take care of minor English errors and we have done this carefully.

We thank all reviewers, especially Reviewer 1, for their constructive comments and their attention to detail which greatly improved the quality of this paper.

Reviewer 3 Report

To the authors, 

Well done on the extensive changes you have made to this manuscript. I think this has really improved the clarity of the work and has made the paper a lot stronger. I have made a few, relatively minor, comments on the attached document. 

Author Response

Comments and Suggestions for Authors

To the authors,

Well done on the extensive changes you have made to this manuscript. I think this has really improved the clarity of the work and has made the paper a lot stronger. I have made a few, relatively minor, comments on the attached document.

Response: We thank you for this comment.

Response: We have carefully reviewed and incorporates all minor comments/corrections in the marked up document provided.

Both reviewers suggested that we take care of minor English errors and we have done this carefully.

Sincerely,

Ahmed Al Ali

Sabir Bin Muzaffar

Waleed Hamza

This manuscript is a resubmission of an earlier submission. The following is a list of the peer review reports and author responses from that submission.

Round 1

Reviewer 1 Report

I enjoyed reading this manuscript. This manuscript presents foraging ecology data on Western Reef Heron which is a species not well studied and therefore presents interesting findings. 

My primary concerns are with methodology and overall presentation of results in relation to the title/abstract. In the methods, the authors do not clearly state how they distinguished between juveniles, sub-adults, breeding and non-breeding individuals. While herons can often be grouped into juveniles and adults, I am not familiar with how one would distinguish into these four groups. For example, I am not sure what is difference between juvenile and subadult and not sure the authors state there differences. Also, how are non-breeding and breeding adults distinguished? The second concern is really about the title and abstract which suggest this paper is about foraging differences in morphs but the paper presents more about foraging differences between seasons and between age groups whereas morphs seem to be less emphasized. Furthermore, Fig 2 and 3 might be more interesting if compared foraging behaviors between age classes and/or morphs since that is where some differences were detected. I think the paper could be strengthened by emphasizing the differences where detected, adequately illustrating those differences in tables and figures and focus the discussion on their results. 

Edits by line

15: "statistically" mispelled

18: should be "13 behaviours"

19: Statistical variation is of interest, but was there biological significance to this variation?

21: I would suggest the use of word "influence" versus "impact"

26: gularis is mispelled

26: Should be "An ethogram" versus "As ethogram"

27: recommend "was created based on..."

30: Why not mention morphs here instead of only seasons and age groups?

55: Reference 4 does not include all herons, it is only on Western Reef Herons. Recommend Kushlan and Hefner or Kushlan references that discuss color morphs in Ardeidae

57: Is it appropriate for a sentence to begin with [5]? Seems odd to me

61: Why are the references not in order (i.e. 3, 5, 8?)

67: Should scientific names be in italics?

83: Recommend authors consider Shah et al. 2018 (Waterbirds) as good reference on Western Reef Herons in U.A.E. Seems like a better reference than a bird check list reference. 

103: Should be Western Reef Heron, not Egret

104: Again, recommend influence versus impact

129: Recommend using "each" versus "every"

142: How were the birds randomly chosen? Also, sentence needs to be reworded to state birds randomly chosen, not "randomly bird is selected"

144: preening does not need to be capitalized

146: "recorder" mispelled

157: why not present foraging efficiency of color morphs?

157: again, how were age classes determined?

Table 1: I am not sure this table is needed. If so, it needs to be restructured as all of the data in the columns are truncated, and so the data is not readable

Fig 3 is hard to read with the current color shading. Also, I would recommend that a figure be offered that shows foraging behaviour differences between age groups and/or color morphs

211-214: Is there any quantified data available for these results?

216-223: Seems to be good data to show in ethogram figure since appears to be main subject of the manuscript

241-253: Again, should species names be in italics?

259-261: IS there any references that resources are more available in winter than summer in UAE?

270: Not sure what authors mean by "diurnal increase" in food. This is tidal changes so they dont necessarily occur during day time. Do they mean daily increase?

314: I think the authors should consider water turbidity as possible factor in color morphs differences. If white plumage is more cryptic to aquatic prey, then water turbidity could negate this crypsis. The authors state that UAE waters has high amounts of sediment (increased turbidity), this could partially explain dark to white morph ratios.

330: Reference is incorrect. It is Green, M. C. 

Reviewer 2 Report

A potentially interesting paper on western reef heron behaviour, but the manuscript needs much more work to make it scientifically sound.

There is a very limited period of data collection, only 15 days per two seasons, and from the description of the methods presented, this study would not be able to be replicated. The statistical testing is not well explained and it is hard to work out how data were actually analysed. From the methods presented, how were behavioural data taken from each one-minute sample point?

There are no definitions of what each behaviour is in the ethogram. How can you show the reader what to look for so that they could also identify these behaviours too?

I am unsure how you can calculate behavioural diversity from your data as the type of data collected is not well explained. 

Is a paired sample t-test applicable to these data? Are they all independent data points?

How many egrets were observed overall?

Did you analyse climate / weather information alongside of animal behaviour? How can you say that different behaviour patterns are explained solely by colour? What about age, sex, experience, habitat quality, physiological state, moult sequence, weather and food availability? How were all of these factors controlled for?

The tables are poorly presented and hard to read. Please check the quality of the information presented and the format of the titles. The graph has no values on x and y axes and hence is hard for the reader to infer results from.

There is no overall conclusion provided to summarise the discussion. 

Please re-write this paper, with more data. Increase the amount of data collected across different times of the year and assess the characters of the individual birds more closely so you can factor in other variables that may impact on foraging behaviour, alongside of bird colour.

Reviewer 3 Report

To the authors, 

Here is a summary of my main points:

My main points are around clarity of the methods used. Make sure also that anything you refer to in your results/discussion has been included in the methods. Methods should be clear enough that someone else can repeat them. Make sure your statistical tests are approrpiate for the data you are assessing. I would also recommend adding in a paragraph to your discussion which links your results back to the management plans you referred to in your lay summary. Finally, you need to include a conclusion at the end of your discussion, to wrap up your study and its importance. 

I have also included a document with specific recommendations on sections/lines. I hope this is helpful and useful for you. 

Lay summary and abstract

Check the English in both of these – I have made some recommendations to the simple summary at the end of this document. Hopefully that helps with the abstract also.

Line 14: which information?

Line 16: refer back to this aim in your conclusion

Line 22: expand on this, how can they contribute to these management plans?

Line 28: it is not clear what sampling occurred

Line 37: add in why the study is important – what is the relevance of this new information, why do we need to know it

Introduction

Line 57 (and throughout): state the author not just the number of the reference (e.g. Bates [5] stated…)

Line 80: Add in UAE after first time it is used in full

Line 89: is it just a case of it could vary or does it vary by region?

Line 100: ‘is essential’ rather than ‘are’

Methods

The methods need clarifying in places to make the whole section clearer. Make sure that if you refer to things in the results or discussion that you have detailed how you collected/analysed that data in your methods. Methods should be written such that anyone can pick up your work and go and repeat the procedure.

Line 126: specify which year

Line 130: How were the one minutes chosen? Did you do one minute of sampling then have a 4 min break and so on? What kind of sampling did you use during the one minute of observations? Need to clarify methods – they should be repeatable, so a little more information is needed here

Line 132: remove ‘to’

Line 133 – 140: I would move this to below the information about sampling so that all of the information about your sampling technique is combined together

Line 141: is the count of other birds referring to the number of species present or the total number of birds – clarify

Line 142: add in ‘chosen’ after randomly – how was this bird chosen? what did you do to prevent accidental bias?

Line 150 – 152: This would be better in a table. The ethogram should include the behaviour and a definition of it for clarity. This should be separate from the results

Line 157: How did you define your age groups?

Line 162 – 170: These tests need to be unpaired tests as you are comparing two different populations of birds. Paired tests are only used if you are testing the same population under different conditions. You need to state which actual test you used throughout

Line 169: suggesting changing ‘the value of alpha…’ to ‘the significance value was set at 0.05 for all’

Results

There are some things detailed in the results which weren’t covered in the methods. I have tried to detail these in the following comments but make sure everything you have results for have been detailed in the methods.

Line 179: quantify ‘occasionally’

Line 179 – 180: not clear here, I thought you were only looking at the behaviour of the western reef heron?

Line 181 (and throughout): ‘data not shown’ – where is this data? If you are going to refer to it you need to include it in the paper (or reference another paper if that is where it is detailed).

Line 182 – 183: add in percentages

Figure 3: Some of the colours are a little hard to differentiate. Put a title on the y axis

Line 204: you need to include data that you are referring to

Line 207 – 208: did you test preference with a statistical test on this? If not you can’t say this…

Line 212: I thought your observations didn’t start until 7.30, how do you know birds weren’t feeding before this time?

Line 213 – 214: How are you defining high and low feeding activity?

Line 215: Tidal effect analyses aren’t included in methods

Line 217 – 218: which comparisons?

Line 222: How did you control for the effects of season with this test?

Line 229 – 230: what are the stats across seasons?

Discussion

Make sure things included in here are all things that have been included in your methods and results. Presently there is some information which isn’t covered in either. I think this section would benefit from a short paragraph on how your findings relate to your statement in the lay summary ‘Results can contribute to species management plans and wetland management plans as well.’ I don’t think it is clear enough in the discussion at present. You also need to add a conclusion to the end of your discussion to wrap up the study and its importance.

Line 232: You don’t detail abundance in your results… you only say number of species

Line 238: Is this heron’s? if so it might be useful to reiterate in this summary the numbers of herons in summer vs winter

Line 241: recommend rewriting to say ‘the most frequently recorded behaviours were ‘stand and wait’ and ‘walking slowly’ although a range of other behaviours were observed.

Line 247: Is Hitoshi a reference? If so include number

Line 249 and throughout: latin names in brackets

Line 253: Is this normal for both morphs? In the intro you discuss the fact that the two morphs may have different hunting techniques?

Line 255: reference for first sentence

Line 271 – 272: ‘the finding that…’ I didn’t see this in your results – if this is the case then make sure it is clear in your results

Line 291 – 292: 13.3% and 17.8% - these values aren’t mentioned in your results

Line 295: did you look at the effects of group size? If so then detail how you did it and what your results were earlier in text also

Line 300: types of food aren’t detailed in your methods or results

Rewording of simple summary recommendation

Simple Summary: Western reef heron (a species within the Ardeidae family) is widely spread in its geographical distribution. Yet, feeding behaviors have never been studied for this species. This study of feeding behavior in birds that have two color morphs is vital for understanding the impact of coloration on behavior and interactions with prey. This will help towards understanding more about factors affecting this color variation. We conducted a field survey to collect all the information in two seasons (Summer and Winter) to establish feeding behaviors and analyze feeding behavior in relation to the bird color morph to understand the impact on wetlands as the demand of eco-tourism and wetland protection is increasing globally. Despite the record of 28 feeding behaviors in species from the Ardeidae family, our research showed that the Western Reef heron uses only 13 of the previously described feeding behaviors. Behavioral diversity and feeding efficiency varied between the two morphs and between the two seasons. This suggests that there is an impact of camouflage on feeding behavior. Results can contribute to species management plans and wetland management plans as well.